# “You’ve Got to Pick Your Battles”: A Mixed-Methods Investigation of Physical Activity Counselling and Referral within General Practice

**DOI:** 10.3390/ijerph17207428

**Published:** 2020-10-12

**Authors:** Benjamin J. R. Buckley, Stephanie J. Finnie, Rebecca C. Murphy, Paula M. Watson

**Affiliations:** 1Liverpool Centre for Cardiovascular Science, University of Liverpool, Liverpool L7 8TX, UK; 2Research Institute for Sport and Exercise Sciences, Liverpool John Moores University, Liverpool L3 3AF, UK; R.C.Murphy@ljmu.ac.uk (R.C.M.); p.m.watson@ljmu.ac.uk (P.M.W.); 3Emergency Department, Royal Liverpool University Hospital, Liverpool L7 8XP, UK; stephanie.finnie@liverpoolft.nhs.uk

**Keywords:** primary care, exercise medicine, health promotion, socio-ecological model, mixed-methods

## Abstract

One in four people say they would be more active if advised by a general practitioner (GP), yet 72% of GPs do not discuss physical activity (PA) with patients and 80% of GPs are unaware of the PA guidelines. The aim of this study was therefore to investigate GP perspectives on PA counselling and referral and interpret these within the context of the socio-ecological model (SEM). Fifty-six GPs completed an online survey to investigate factors influencing PA counselling and referral. Semi-structured interviews were then conducted with seven GPs to explore topics in more depth. Interview data were analysed thematically and mapped to the SEM. GPs were more likely to discuss PA with patients if they were physically active themselves (*p* = 0.004). Influences on PA counselling and referral were identified at the policy (provision of education, priority), organisational (feedback, e-referral), interpersonal (PA as everybody’s business, patient factors) and intrapersonal (knowledge, GP PA levels) levels of the SEM. Multi-level strategies are required to help GPs promote PA and make use of exercise referral schemes, including making PA a strategic priority, introducing systems for feedback from referrals, and involving other members of the care team in PA counselling and referral.

## 1. Introduction

Insufficient physical activity (PA) is the fourth leading cause of global premature mortality [1] and contributes to more than a quarter of deaths in the UK [2]. In 2019, 40% of 40–60-year-olds self-reported walking < 10 min continuously each month at a brisk pace [3]. Interventions to increase PA levels in at-risk populations are therefore of great public health interest. Integrating PA promotion into primary healthcare has been termed a “best buy” for decreasing the burden of non-communicable diseases and improving quality of life [1,4].

Exercise referral schemes are common initiatives throughout the UK, which aim to increase PA levels over 12–26 weeks [5]. They are multi-level interventions that require a joined-up transition from referral to intervention delivery with the aim of long-term PA behaviour change. There is little evidence of consistent PA promotion within primary care, though brief interventions and exercise referral schemes are the two dominant approaches [6]. General practitioners (GPs) are an integral component of the referral system, having regular contact with at-risk patients from diverse socio-economic backgrounds [7]. GPs or family doctors are medical doctors who treat acute and chronic illnesses and provide preventive care and health education to patients. One in four people say they would be more active if advised by a GP or nurse [4]. Despite this, recent work found 80% of GPs in England are unfamiliar with the national PA guidelines, 72% do not discuss PA with patients, 43% were “somewhat confident” in discussing PA with patients, and 55% of GPs reported having no training in providing PA counselling [8].

Due to the complex nature of PA behaviour change, a single intervention component is unlikely to have lasting effects [9,10]. Consideration of the complex nature of physical inactivity and the multiple interacting levels of influence calls for a multi-level approach. According to Bronfenbrenner’s socio-ecological model (SEM) [11], behaviour is a product of intrapersonal, interpersonal, organisational, and policy factors that interact among one another [12,13]. To date, research has focussed on the intervention delivery aspect of GP referral schemes [14]. Recent work from our group demonstrated the promising potential of a co-created exercise referral scheme to improve patient health [15,16,17]. Anecdotal evidence from this work, however, highlighted a lack of communication between GP practices and the referral scheme, and unsystematic referral procedures across different GP practices. The synergistic and multisectoral action of several stakeholders, especially healthcare professionals, is therefore needed to overcome physical inactivity in a sustainable way [6]. Though previous work has investigated health professional perspectives of PA promotion [18], exclusive focus on GP perspectives of how PA counselling and exercise referral can be facilitated has not been previously explored. This sequential explanatory mixed-methods study therefore aimed to investigate GP perspectives on PA counselling and referral and interpret these within the context of the SEM.

## 2. Materials and Methods

Design and setting. This sequential explanatory mixed-methods design study [19] involved an online survey followed by interviews with GPs in Liverpool, UK. Liverpool is the third most deprived local authority in England and has the second highest proportion of Lower Super Output Areas in the most deprived 10% nationally [20]. Data collection occurred from May 2019 to September 2019.

Participants and recruitment. An online survey was distributed to all GPs in Liverpool (*n* = ~350) via the Clinical Commissioning Group (CCG) internal mailing system, bi-weekly news bulletin, social media, and word of mouth. GPs who completed the survey were able to opt-in to take part in a semi-structured interview.

Inclusion criteria. GP partners, salaried GPs, GP registrars, and locum GPs currently working within a surgery in Liverpool were invited to take part. 

Online survey. A ~5-min online survey was distributed (Appendix A) and included questions on demographics, frequency of PA counselling, awareness of the local exercise referral scheme and other PA initiatives, facilitators and barriers to referral, views on who should refer, e-referral, and ideas for improving GP engagement with PA promotion. GPs were also asked about their awareness and use of the “Moving Medicine” resource (part of the national Moving Healthcare Professionals Programme) [21]. Items were presented in a multiple-choice format, where some questions permitted more than one answer, as well as the opportunity to provide qualitative comments. Once a draft survey was developed, it was reviewed by a GP representative and local commissioners to ensure topic relevance. 

Qualitative interviews. Semi-structured interviews were conducted by the first author at the GP’s place of work or via telephone. Interviews lasted an average of 44 min (21 to 119 min) and explored the same topics as the online survey. 

Analysis. Survey data were analysed descriptively and reported as frequencies and percentages. For continuous dependent data (GP PA levels), data were checked for normality and independent T-Tests were conducted to compare PA levels between GPs who discussed PA with patients “some of the time” with GPs who discussed PA with patients “most of the time”. T-Tests were also used to compare GP PA levels between GPs who referred patients for exercise and those that did not. Chi-Square tests were conducted to investigate differences in GP referral and PA promotion behaviours by surgery deprivation. 

Interview data were transcribed and imported into NVivo version 12. A thematic analysis [22] was carried out by the first author, which involved coding, identifying and defining themes, then mapping these themes against the SEM. This included policy level factors, organisational level factors, interpersonal level factors, and intrapersonal level factors. All authors independently analysed three interview transcripts and engaged in regular discussion to continuously refine themes until a best-fit of the data was reached. This process has been recommended to improve rigour and reduce researcher bias in qualitative research [23].

### Ethics Approval and Consent to Participate

NHS ethical approval was granted (REC: 19/HRA/2620—Project: 264222) and written informed consent provided by all participants (implied for the survey).

## 3. Results

Participant demographic data are presented in Table 1, survey responses in Table 2, and interview findings in Table 3. In total, 56 GPs completed the survey (16% response rate) and 7 GPs consented to take part and completed an in-depth interview (1 via telephone).

### 3.1. Survey Responses

Survey responses are presented in Table 2. Seventy-seven percent of GPs asked patients about their PA levels “some of the time” and made 1–5 exercise referrals per month. The most commonly reported challenges for making exercise referrals were a lack of time within consultations (61%), a lack of motivation (39%), and a lack of knowledge about initiatives or how to refer (25%). The most commonly reported facilitators for making more referrals were an e-referral system (68%), clearer marketing information about content (46%), and eligibility (45%). 

GPs who self-reported asking patients about PA “most of the time” were more active (4.7 ± 1.1 days of achieving ≥ 30 min of moderate intensity PA) compared to GPs who asked patients “some of the time” (3 ± 1.7 days of achieving ≥ 30 min of moderate intensity PA; *p* = 0.004). Similarly, GPs who made 1–5 or 5–10 exercise referrals per month were more active (3.7 ± 1.7 days of achieving ≥30 min of moderate intensity PA) than GPs who did not refer patients for exercise (1.9 ± 1.1 days of achieving ≥30 min of moderate intensity PA; *p* = 0.006). When comparing the least and most deprived GP surgeries, there were no significant differences in how often GPs promoted PA to patients (*p* = 0.197) or referred patients for exercise (*p* = 0.370).

### 3.2. Interview Responses

Interview responses are presented in Table 3. Though some GPs noted regularly discussing PA, others highlighted difficulty doing so within a time-limited 10-min consultation. Referral behaviours were mixed, with some GPs making exercise referrals more frequently than others. Several GPs (or their patients) viewed the current exercise referral scheme as “cheap gym access”, whereby the subsidised scheme was considered a means of gaining affordable access to the gym for participants who faced cost barriers. All GPs commented on how exercise referral should be more holistic by incorporating activities outside of a leisure centre environment. It was perceived that this would reduce barriers associated with cost and might encourage more patients to initiate and sustain PA.

Factors perceived to influence PA counselling and referral were identified at the policy, organisational, interpersonal, and intrapersonal levels of the SEM. 

Policy level factors included provision of education and priority of PA. Whilst two GPs had attended an education event on exercise medicine, most GPs identified a lack of education provision both with regards to PA counselling and the local exercise referral scheme. All GPs noted priority as a key factor in whether to promote PA within consultations. For example, several GPs noted that a PA-related Quality Outcome Framework (QOF) target (QOF targets define strategic priorities for GPs in England) would increase the likelihood of them discussing PA and referring for exercise. Other GPs noted that PA promotion is not always the priority within a consultation.

Organisational level factors included a lack of feedback and e-referral. All GPs noted that there was no mechanism for exercise referral scheme providers to feedback to GPs on patient progress. GPs said it would be beneficial to receive patient feedback (such as attendance and outcomes) and noted evidence of successful outcomes could be a helpful motivator when referring future patients to the scheme. There was consensus that an e-referral system would be a welcome improvement to the current paper form referral process.

Interpersonal level factors included PA as everybody’s business and individual patient factors. Participants felt “work is always being dumped on the GP”, yet noted PA promotion is not the sole responsibility of GPs and that other healthcare practitioners may have more time to spend with patients and may be better placed to provide PA counselling and referral. Most GPs mentioned social prescription (an initiative that enables primary care professionals to refer people to a range of local, non-clinical services) [24] as a potentially better model to embed PA promotion into primary care.

GPs noted how it was important to tailor referrals to patients’ circumstances, noting they were more likely to refer patients of a lower socio-economic status for exercise (who might benefit from the subsidised cost), and that it was important to take into account the patient’s mental health status when making referrals. It was noted that PA counselling sometimes felt inappropriate if patients were faced with multiple health issues, and sometimes you had to “pick your battles” instead of asking patients to change several health behaviours at once.

Intrapersonal level factors included knowledge and GP PA levels. GP understanding of PA and the local exercise referral scheme was mixed, with some GPs having attended local information events which encouraged them to refer more often. All GPs noted that their own PA levels affected their promotion of PA with patients; one GP noted that when they used to be unfit and inactive themselves, they did not feel confident talking to patients about improving their PA levels.

## 4. Discussion

In general, GPs noted a lack of education provision for promoting PA and referring patients for exercise. All GPs were clear that “PA is everybody’s business”, i.e., there is over-reliance on GPs to promote PA and other health professionals may even be better suited to do this. We also found that more physically active GPs were more likely to promote PA and refer patients for exercise.

Our findings demonstrate that GPs feel PA promotion is “being dumped on the GP” and GPs strongly advocate that PA counselling and exercise referral is not the sole responsibility of doctors. The concept of exercise as everybody’s business has been previously highlighted [25]. Indeed, more of a “shared care” pathway has been proposed, including physicians, healthcare and exercise professionals that create a community of exercise medicine advocates [26]. The “clearance to exercise” model may have compounded the over-reliance on GPs, by historically relying on them to “sign patients off” for exercise [27].

A cross-sectional study surveyed >500 GPs in Australia and demonstrated that 98% of GPs identified “discussing ways to increase PA” with patients as part of their role, with 53% discussing PA with >10 patients per week [28]. In England, however, 80% of GPs were unfamiliar with the national PA guidelines and only 43% were “somewhat confident” in discussing PA with patients [8]. In agreement with Chatterjee et al. [8], our findings demonstrated that 77% of GPs reported discussing PA during “some consultations” and making 1–5 exercise referrals per month. It is unclear exactly why Australian and English GPs differ in practices/views on PA counselling and referral. The literature implies that a lack of skills, knowledge, and confidence in promoting PA may be due to a lack of formal training and education opportunities [29]. For example, >60% of GPs in Australia were aware of the PA guidelines compared to only 13% in the UK [7]. The most commonly reported facilitators to increasing exercise referrals in primary care, identified by GPs in the present study, included the involvement of other health professionals and better provision of PA education. 

Previous work has discussed the need to embed exercise medicine education into the medical curriculum, with only 56% of UK medical schools including the Chief Medical Officer’s PA guidance in the curriculum [30] and medical students under-appreciating the risk of physical inactivity on mortality [31]. The Moving Medicine initiative was developed to provide resources to support PA promotion in primary care across England as part of the Moving Healthcare Professionals Programme [21]. Such resources may help improve knowledge of exercise medicine and PA counselling. Brannan et al. [21] reported some encouraging progress, with 74% of medical schools agreeing to implement PA-related modules into the curriculum and reaching >17,000 healthcare professionals via postgraduate education provision. Despite such promising statistics, our findings demonstrate that 15% of GPs were aware of Moving Medicine and only 5% reported its use. Correspondingly, GPs in the present study called for improved education provision, more hard copy resources, and exercise medicine education events in the local area. Promisingly, the Royal College of General Practitioners [32] have released a toolkit to help GP practices achieve “Active Practice status” by advocating PA conversations with patients. It is therefore critical that such resources are made available and accessible to health professionals “on the ground”.

Finally, one example of a “top down” approach to increase PA promotion is a PA-based QOF. The QOF is a principle target-based system through which GPs are incentivised to prioritise certain aspects of care. Previous work has demonstrated that the inclusion of health parameters within QOFs can lead to improved care [33]. Our findings support that of Savill et al. [7], who argued that in an increasingly target-based healthcare system, incorporating PA-based QOFs represents a promising opportunity to promote PA for health. All GPs in the present study noted that PA promotion needed to be more of a priority within primary care, with most GPs suggesting a PA-related QOF target as a solution. 

### Methodological Considerations

This is the first known study to focus exclusively on GP perspectives of PA counselling and exercise referral, using a mixed-methods design. Undertaking a sequential explanatory mixed-methods design, this work provides rich data regarding current practices and importantly, GP perspectives on how to improve provision of PA promotion. A junior doctor working in general practice was recruited as a co-author, providing a frontline lens to the research. Finally, the qualitative data were mapped to the SEM to provide multi-level recommendations to improve GP provision of PA promotion. 

A number of limitations warrant acknowledgement. First, this study focussed solely on GPs, since in the study location the majority of exercise referrals were provided by GPs. In other locations in the UK this varies however, and future research would benefit from understanding the perspectives of other healthcare professionals (e.g., nurses, physiotherapists) who are well-placed to advocate PA within their practice. Second, as GPs in this study were self-selected, it is likely some participants had a personal interest in PA, which needs to be considered when interpreting the findings. Third, to maintain survey participant anonymity, we were not able to link GPs who were interviewed directly to their survey responses, which limited the ability to evaluate representation of the survey participants. Fourth, the low survey response rate (16%) and inability to compare our data to non-responders means the generalisability of these findings is unclear. Finally, we did not observe a proportionate use of referrals according to local deprivation, which is in contrast to previous research [34]. As 96% of GPs in the present study worked within the most deprived areas however, it is likely this affected our ability to observe any relationships between referral behaviour and deprivation. 

## 5. Conclusions

The promotion of PA and exercise is an urgent public health agenda. However, findings from the present paper highlight that GPs, although well placed to promote exercise as medicine, are not sufficiently educated or supported to do so, and feel like work is being “dumped” on them. This work highlights that several multi-level strategies are required to help GPs promote exercise as medicine through PA counselling and use of exercise referral schemes. These strategies involve policy (make PA a strategic priority), organisation (introduce systems for feedback from referrals), interpersonal (involve other members of the healthcare team in PA counselling and referral), and intrapersonal (improve GP knowledge of exercise as medicine and available resources both online, and in the community) level factors that can be targeted. Finally, implications for practice and research are presented below.

### 5.1. Implications for Practice

GPs feel like PA promotion is being “dumped on them” and more of a collaborative effort with health and exercise professionals is needed to promote exercise medicine.The Moving Medicine initiative is a high-quality resource to support PA promotion in primary care as part of the Moving Healthcare Professionals Programme. Better marketing, education, and integration of such resources into primary care is needed where it can have the greatest public health impact.PA promotion and exercise medicine needs to be made a key priority within primary care. This may include improved undergraduate training, education provision for existing health professionals, better links with exercise professionals, and consideration of PA-related QOFs, as suggested by GPs in the present study.Moving from paper-based to electronic systems may facilitate exercise referrals in primary care.

### 5.2. Implications for Research

Advocation of multi-level approaches (e.g., socio-ecological model) when evaluating complex interventions, specifically those relating to PA promotion and exercise medicine.As social prescription (which enables primary care professionals to refer patients to a range of local, non-clinical services) was stated by most GPs as a promising model to embed PA into primary care, evaluation of social prescription is needed to determine: (a) what a social prescribing model looks like in practice, (b) what staff competencies are optimum for social prescribing, and (c) the impact social prescribing has on PA promotion.

## Figures and Tables

**Table 1 ijerph-17-07428-t001:** Participant demographic data provided by GPs in Liverpool, UK.

Survey Participant Demographics (*n* = 56)	
Age (years)	45.5 ± 11.3
Sex (% Female)	28 (50%)
Ethnicity (% White British)	43 (77%)
Years worked as a GP	16 ± 11
Full-time	29 (52%)
Average surgery Index of Multiple Deprivation ^#^	2 ± 2
GP activity levels Meeting the PA recommendations *	3.4 ± 1.8 days/week of ≥30 min of MVPA 13 (23%)
Interview Participant Demographics (*n* = 7)	
Age (years)	41.9 ± 12.2
Sex (% Female)	2 (29%)
Ethnicity (% White British)	5 (71%)
Years worked as a GP	14 ± 12
Full-time	6 (86%)
Average surgery Index of Multiple Deprivation ^#^	3 ± 2

GP, General Practitioner; PA, Physical Activity; MVPA, Moderate-to-Vigorous intensity Physical Activity; * Chief Medical Officers’ 2019 physical activity guidelines: 150 minutes of moderate-intensity physical activity per week; ^#^ Average GP surgery Index of Multiple Deprivation (accessed January 2020); The Index of Multiple Deprivation combines information from seven domains to produce an overall relative measure of deprivation. Presented here in deciles, on a scale from 1 (most deprived 10%) to 10 (least deprived 10%) of England. Most GP surgeries (96%) were within deprived areas (decile 1–4).

**Table 2 ijerph-17-07428-t002:** GP survey responses (*n* = 56) regarding physical activity promotion in Liverpool, UK.

GP Referral Statistics	
How often did GPs ask patients about their PA levels	Every consultation = 1 (2%) Most consultations = 12 (21%) Some consultations = 43 (77%) Never = 0 Other = 0
How many referrals did GPs make to the local exercise referral scheme per month	0 = 11 (20%) 1–5 = 43 (77%) 5–10 = 2 (4%) >10 = 0
GP referral to other PA initiatives *(Multiple answers allowed)*	Fitness centres (other than local exercise referral scheme) = 22 (39%) Parkrun = 23 (41%) Forever active (local fitness initiative targeted at over-50s) = 4 (7%) Other 13 = (23%) Summary of qualitative responses: Walking/cycling groups. Health trainer (health behaviour support and link to local initiatives). Weight management. Yoga. Tai chi. Liverpool Football Club Foundation (local charity providing health and wellbeing support through PA) No response = 15 (27%)
**Facilitators and barriers to referral**	
What prevents GPs from making referrals to the exercise referral scheme? *(Multiple answers allowed)*	Lack of motivation = 22 (39%) Lack of knowledge about initiatives or how to refer = 14 (25%) Lack of belief in effectiveness = 1 (2%) Lack of time within consultations (other priorities) = 34 (61%) Personally don’t believe exercise is an effective treatment = 0 Other = 9 (16%) Summary of qualitative responses: Not always relevant. Patient unable to access facilities. Practice nurse makes referrals (*n*=3). Unclear referral pathway. Paper referral form asks unnecessary questions. No response = 2 (4%)
What would encourage GPs to refer an exercise referral scheme? *(Multiple answers allowed)*	Electronic referral system = 38 (68%) Clearer marketing information about content = 26 (46%) Clearer marketing information about eligibility = 25 (45%) If the scheme was more physical activity focused, rather than only based at fitness centres = 17 (30%) Better feedback about patient progress = 15 (27%) Knowledge of effectiveness = 8 (14%) Other = 10 (18%) Summary of qualitative responses: Referral system needs to be quick and easy, shouldn’t need patient data like height and weight or pulse. Self-referral (*n*=6). Role for non-clinical support workers. No response = 1 (2%)
Most effective ways to communicate with GPs *(Multiple answers allowed)*	Hard copy leaflets = 34 (61%) Hard copy posters = 30 (54%) Leaflets for referrers with exercise referral scheme information on = 23 (41%) Short instructional video = 10 (18%) Exercise referral scheme representatives to present details at GP forums = 16 (29%) An additional event for GPs = 2 (4%) Online circulation of initiative information = 34 (61%) Other = 3 (5%) Summary of qualitative responses: Receptionists and other non-clinical staff involvement (GPs not the only people in contact with patients). No response = 4 (7%)
**Moving Medicine engagement**	
GP awareness of Moving Medicine initiative	8 (14%)
GP use of Moving Medicine initiative	3 (5%)

**Parkrun**, free, weekly 5km running events with over 700 locations globally; **Forever active**, local over 50s exercise programme; **Moving Medicine**, An initiative which provides resources to support PA promotion in primary care as part of the Moving Healthcare Professionals Programme set up by Public Health England and Sport England [21].

**Table 3 ijerph-17-07428-t003:** Summary of factors influencing GP PA promotion embedded within the SEM: Qualitative interview findings.

SEM Level *	Themes	Example Quotes
**Policy level**-Factors affected by national policy.		
	Provision of education	“you need to increase the doctors’ understanding so they’re more willing to say... Don’t reach for your prescribing pad, as it was, reach for your exercise referral sheets” (P1,M,39). “Most doctors are very willing to learn new things, especially if they can get a CPD point out there” (P1,M,39).
	Priority	“… trying to shoehorn that [PA counselling/referral] in your ten-minute consultation… I suppose, yes, we’re doing it [exercise referral] on an ad hoc basis if we’ve got time to put it into consultation, and sometimes you think, "Oh yes, I could have mentioned the scheme but didn’t have enough time to today” (P2,M,36). We get people coming in asking about three things at once, and then it’s trying to prioritise, and unfortunately again, either GPs or the patients sometimes push that [PA counselling/referral] lower down the list than it should be” (P3,F,53).
**Organisational level**-Factors related to resources, programmes, and services.		
	Feedback	“I’ve never had a document [from the exercise referral scheme], I’ve never had a phone call, I’ve never had anything at all” (P6,M,44).
	E-referral system	“I think we’ve got to move towards electronic, haven’t we? If it self-populates with pulse, blood pressure, height, weight, any other information, medical conditions that are relevant, it would be a lot easier” (P3,F,53).
**Interpersonal level**-Factors related to GP interaction with others, including factors that are “intrapersonal” to the patient.		
	PA as everybody’s business	“… from hospital doctors to people in the gym, they’re all going to say “go to your GP”, and most of the time they’re not the person that they should be going to, and I think we feel like work is always being dumped on the GP” (P7,F,31).
	Individual patient factors	“You sort of have to tailor it to the patient’s personality. For example, if they are depressed, then the motivation to go out of the front door isn’t there, and some people just never had that habit of doing exercise, so trying to work out a way to just introduce it, like a ten-minute walk or… inside the house maybe to start with, and then maybe trying to build it up from there to doing this [PA] referral scheme” (P2,M,36). “sometimes you’ve got to pick your battles and go with what’s the most important thing, what does the patient want from that consultation on that day” (P7,F,31).
**Intrapersonal level**-Factors related to solely GPs at the individual level.		
	Knowledge	“I didn’t have as good an understanding until I went to the re-launch of the Exercise for Health [local exercise referral scheme], they had lots of speakers, [which] sort of re-invigorated me referring people” (P3,F,53).
	GP PA levels/personal interest	“I have a personal interest in it [exercise], and therefore I’m a bit more motivated to know a bit more and promote that as a treatment in its own right. So probably my enthusiasm for referring, and therefore my understanding, I would suspect, is probably a bit more than some of the others [GPs]” (P1,M,39).

* SEM level definitions have been created specifically for this study and underpinned by Sallis, Owen, & Fisher [13]. Interview quotes are presented with the participant number, sex, and age of the interviewee e.g. (P1,F,45) = Participant 1, Female GP, 45 years old; GP, General Practitioner; PA, Physical activity; SEM, Socio-Ecological Model.

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
