# Peer review of "“You’ve Got to Pick Your Battles”: A Mixed-Methods Investigation of Physical Activity Counselling and Referral within General Practice"

_ijerph, 2020, doi:10.3390/ijerph17207428_

Round 1
Reviewer 1 Report
This mixed methods study aimed to “…investigate GP perspectives on PA counseling and referral and interpret these within the context of the Socio-Ecological Model (SEM).” The paper presents various strengths, including identification of a clear gap in the literature and the opportunity to enhance further healthcare-provider physical activity referral strategies with greater input and ‘co-construction’ with general practitioners; the overall mixed methods approach (sequential design)- with good specificity overall with describing methods- as well as inclusion of methods to enhance rigor (e.g., all co-authors involved in coding data); and good organization, relevant citation of literature, and insightful findings and discussion. These strengths notwithstanding, some facets of the paper may benefit from further enhancement:
- Sample: Can the authors clarify how many physicians opted in to the in-depth interviews – in addition to how many completed the interviews? Also, can the authors provide more descriptive characteristics of the 7 GPs who opted to be interviewed in order to compare with the larger sample of GPs who completed the survey? Lastly, while this reviewer appreciates the inclusion of a response rate, can they clarify how they determined the denominator for that response rate?
- Methods: This reviewer appreciated the mixed methods design of the study. While the authors appropriately note this was a sequential design, they may consider further specificity by stating that this was a sequential explanatory design (quantitative to qualitative)- based on this reviewer's understanding of their approach (e.g. (Halcomb and Hickman, 2015; Shorten and Smith, 2017; Creswell and Plano Clark 2011).
- Results:
- The authors state: “There were no trends in GP surgery deprivation and how often GPs promoted PA and made referrals or GP PA levels and 108 frequency of exercise referrals (p>0.05)." (lines 107-109). As ‘trends’ is often used in relation to time, and this was a cross-sectional survey, I recommend other language use (e.g., “there were no significant differences in GP surgery deprivation scores by how often GPs promoted PA with their patients”- or something to that effect.
- Lines 114-115: Most GPs (or their patients) viewed the current exercise referral scheme as “cheap gym access”, and all GPs commented how exercise referral should be more holistic by incorporating activities outside of a leisure centre environment.” Can the authors provide more context regarding ‘cheap gym access’?
- General comment on Tables: Consider including more descriptive titles so that ‘tables stand on their own’ for interpretation (e.g., include place, date, population clearly specified, possibly name of study).
- Lines 202-204: “This is the first known study to represent the perspectives of GPs on PA counseling and exercise referral.” This statement is a bit puzzling for this reviewer given that the authors previously cite data on GP and PA referrals from Australia, not to mention previous review studies on primary health care professionals’ physical activity promotion behaviors (e.g. Huijg et al., 2015) and other studies (Leijon et al., 2009; Crone et al., 2003; James et al, 2007; among others). Can the authors contextualize further this statement? (perhaps this is the first study to focus only on GPs, or to use a mixed methods approach?).
- The authors appropriately acknowledge relevant limitations of the study, which this reviewer appreciates. One limitation not mentioned was the low response rate of 16%, which merits acknowledgement in terms of the questionable generalizability of the findings across GPs.
- Discussion: This reviewer found the discussion to be rich and relevant in exploring key findings from the study. Nice work.
- Implications for research: The authors state: “Evaluation of social prescription to determine what it is and the impact it has on PA promotion in primary care." this reviewer recommends further context to better understand this implication and how it relates to the findings.
Reviewer 2 Report
Promoting PA through GP referral and counselling should be more acceptable to the high risk group. This study aimed to investigate GP’s perspectives on PA counselling and referral and interpret these within the context of the Socio-Ecological Model. The idea and theory of the study is interesting, but the design and implementation had some problems which weakened the findings and impacts of the study. Below are some of my considerations:
- The response of survey was low. The authors should give a comparative analysis in table 1 to address whether those participating GPs were comparable to those not participated in key characteristics, such as years of work, age, gender, etc.
- The statistics of the finding on “GP being physically active themselves or not (p=.004)” seems problematic. The authors said that they used T-test. Is it student-t test or not? To me, this result could only come from Chi-Square test. The statistical analysis was very weak in this study.
- The study aimed to give a multifactorial/multilevel analysis to address the research question. However, even applied both quantitative and qualitative studies, the explanations were more from the PA referral scheme’s point of view rather than from the GP’s perspectives, such as motivation, performance evaluation, workloads and their own perceptions on PA.
- The topic migh be more suitable for domestic journals.
Reviewer 3 Report
Thanks for the opportunity to review this paper. Generally, this is a novel paper that is suggested as a potential accepted paper. By screening the paper, I have some comments for the authors’ considerations.
Point 1: Line 40: define the full expression of GPs in the main body of the text, even though the GPs has appeared in the abstract section.
Point 2: I think the introduction was not sufficient, I suggest adding more relevant contents. Please consider it. For example, what is GPs? How the effectiveness of GPs to increase the level of PA across the previous literature? What are the current gaps of GPs to increase PA? it is very important to incorporate those into introduction to support your study rationale.
Point 3: could please reconstruct the method section, its current presenting format seem unfriendly. Please show the method section more clearly and increase the reading flow.
Point 4: how did you promise the reliability and validity of the this.
Point 5: in the analysis, you mentioned that “Survey data were analysed descriptively and reported as frequencies and percentages. T-tests were conducted to investigate differences in GP PA levels and referral behaviours”. But I did not read the description on assess GP PA levels and referral behaviours? Could you please clarify this issue?
Point 6: in the results section, you mentioned policy level factors, organisational level factors, Interpersonal level factors, Intrapersonal level factors, but I did not see the related measurement in the method sections. Please clarify it.
Round 2
Reviewer 2 Report
The authors have revised the manuscript accordingly and improved to a large extent. The statistics are described clearly now.
The low response is the main weakness of the study which affects the validity of the study. The authors have given their efforts to address the problem through qualitative study, and the limitations of study have been presented in the discussion.
Reviewer 3 Report
The authors have addressed the comments